# Evaluation of Antidepressive-like Behaviours and Oxidative Stress Parameters in Mice Receiving Imipramine-Zinc Complex Compound

**DOI:** 10.3390/ijms241814157

**Published:** 2023-09-15

**Authors:** Aleksandra Szopa, Mariola Herbet, Ewa Poleszak, Anna Serefko, Agnieszka Czylkowska, Iwona Piątkowska-Chmiel, Kamila Kasperek, Andrzej Wróbel, Paulina Prewencka, Bernadeta Szewczyk

**Affiliations:** 1Department of Clinical Pharmacy and Pharmaceutical Care, Faculty of Pharmacy, Medical University of Lublin, 1 Chodźki Street, 20-093 Lublin, Poland; aleksandra.szopa@umlub.pl (A.S.); anna.serefko@umlub.pl (A.S.); 2Department of Toxicology, Medical University of Lublin, 8 Chodźki Street, 20-093 Lublin, Poland; mariola.herbet@umlub.pl (M.H.); iwona.piatkowska-chmiel@umlub.pl (I.P.-C.); kamila.kasperek.pharm@gmail.com (K.K.); 3Laboratory of Preclinical Testing, Chair and Department of Applied and Social Pharmacy, Medical University of Lublin, 1 Chodźki Street, 20-093 Lublin, Poland; ewa.poleszak@umlub.pl; 4Institute of General and Ecological Chemistry, Faculty of Chemistry, Lodz University of Technology, Zeromskiego 116, 90-924 Łódź, Poland; agnieszka.czylkowska@p.lodz.pl; 5Second Department of Gynecology, Medical University of Lublin, 8 Jaczewskiego Street, 20-090 Lublin, Poland; wrobelandrzej@yahoo.com; 6Scientific Circle, Department of Clinical Pharmacy and Pharmaceutical Care, Medical University of Lublin, 1 Chodźki Street, 20-093 Lublin, Poland; paulina.wojtowicz35@gmail.com; 7Department of Neurobiology, Maj Institute of Pharmacology, Polish Academy of Sciences, 12 Smętna Street, 31-343 Kraków, Poland

**Keywords:** imipramine-zinc complex, antidepressant-like activity, mice, oxidative stress, glutathione peroxidase, glutathione reductase, total antioxidant status, environmental stress

## Abstract

The study aimed to evaluate the antidepressant-like effects of an imipramine-zinc (IMI-Zn) complex compound on mice and assess the level of oxidative stress parameters. The research also investigated whether the IMI-Zn complex showed superior antidepressant activity compared to individual treatments of both compounds at effective doses and their joint administration at subtherapeutic doses. The study was conducted on mice. Forced swim (FST), tail suspension (TST), and locomotor activity tests were used for behavioral studies. The results demonstrated the IMI-Zn complex’s dose-dependent antidepressant potential when orally administered to mice. Its efficacy was similar to the separate administration of therapeutic doses of imipramine (IMI) and zinc (Zn) and their joint administration at subtherapeutic doses. Moreover, subjecting mice to acute stress did not significantly affect the activity of on glutathione peroxidase (GPX), glutathione reductase (GR), and total antioxidant status (TAS), possibly due to the short exposure time to the stress stimulus. By developing the IMI-Zn complex, it might be possible to simplify the treatment approach, potentially improving patient compliance by combining the therapeutic effects of both IMI and Zn within a single compound, thus addressing one of the contributing factors to non-compliance in depression therapy. The IMI-Zn complex could be a valuable strategy to optimize therapeutic outcomes and balance efficacy and tolerability.

## 1. Introduction

Depression is a complex and debilitating mental health disorder affecting millions worldwide [1,2]. It is characterized by persistent sadness, loss of interest or pleasure, disturbed sleep, altered appetite, and impaired cognitive function [3,4,5,6,7]. The pathophysiology of depression involves multifactorial mechanisms, including dysregulation of neurotransmitters, impaired synaptic plasticity, and increased oxidative stress (for review, see [8,9,10,11,12]).

Conventional antidepressant drugs, such as selective serotonin reuptake inhibitors (SSRIs) and tricyclic antidepressants (TCAs), have been the mainstay of treatment for depression. Imipramine (IMI), a prototypical TCA, has demonstrated efficacy in ameliorating depressive symptoms [13,14]. However, the therapeutic response to IMI varies among individuals, and some patients may experience adverse side effects, highlighting the need for alternative or adjunctive treatment options [15,16]. Currently, available therapeutic strategies for depression are very often ineffective or not very effective. Perhaps one of the reasons is that their mechanisms of action do not consider the multifactorial basis of the disease. Therefore, it is essential to search for new therapeutic strategies for depression based on several mechanisms of its pathogenesis. The antioxidant potential seems to be particularly important in this aspect.

In recent years, there has been growing interest in the role of micronutrients, particularly zinc (Zn), in mental health disorders (for review, see [17,18]). Zn is an essential trace element in various physiological processes, including neurotransmission, neurogenesis, and antioxidant defense mechanisms [19,20,21,22,23]. Several studies have reported altered Zn levels in individuals with depression, suggesting a potential link between Zn deficiency and depressive symptoms [24,25,26,27,28,29].

Moreover, Zn supplementation has shown promising results in preclinical and clinical studies as adjunctive therapy for depression [18,27,28,29,30]. Zn has been implicated in the regulation of brain-derived neurotrophic factor (BDNF) signaling [28,31,32,33,34,35,36], which plays a crucial role in neuroplasticity and neuronal survival [37]. Additionally, Zn exerts antioxidant effects by modulating oxidative stress markers, including superoxide dismutase (SOD), catalase (CAT), and malondialdehyde (MDA), thereby attenuating oxidative damage in the brain [38].

Given the evidence supporting the involvement of Zn in depression and its potential therapeutic benefits, combining Zn with conventional antidepressant drugs may offer a novel approach to enhance treatment outcomes. Therefore, this study aims to assess the effects of an imipramine-zinc (IMI-Zn) complex compound on depressive-like behaviors and oxidative stress parameters in mice. Additionally, presented research was scheduled to determine whether the IMI-Zn complex exhibits superior antidepressant-like activity compared to individual treatments at effective doses and their joint administration at subtherapeutic doses. Furthermore, the study aims to elucidate the possible mechanisms underlying the observed effects by examining relevant biochemical and neurochemical parameters. By evaluating both behavioral and biochemical aspects, the study seeks to provide insights into the potential therapeutic value of the IMI-Zn complex compound as a treatment option for depression. Ultimately, this research may contribute to developing more effective and safer treatment strategies for depression, addressing the unmet needs in mental health care. The study also aimed to evaluate oxidative stress parameters in mice’s blood treated with the IMI-Zn complex. The IMI-Zn complex compound is a novelty in research into new therapeutic approaches to treating depression. This study determined the effect of the IMI-Zn complex on glutathione peroxidase (GPX), glutathione reductase (GR), and total antioxidant status (TAS) in mice subjected to acute environmental stress. The stress inducer in mice was the forced swimming test that the mice underwent in the FST (forced swimming test). FST can lead to the formation of oxygen free radicals and increased oxidative stress [39,40].

## 2. Results

### 2.1. Behavioural Examination

*Dose-effect studies*. According to the findings presented in Figure 1, the IMI-Zn complex exhibited antidepressant-like activity in the FST (Figure 1A) and TST (Figure 1B) in mice when administered orally. The antidepressant-like effect was significant within the 20–100 mg/kg dose range. The statistical analysis using one-way ANOVA revealed significant results for both the FST [F(7,72) = 14.10; *p* < 0.0001] and the TST [F(7,72) = 9.362, *p* < 0.0001]. Furthermore, the positive control, IMI, administered at a dose of 90 mg/kg, also significantly reduced the immobility time in both behavioral tests, confirming the validity of the experimental approach.

Notably, the tested doses of the IMI-Zn complex did not influence locomotor activity. Only the group receiving IMI at a dose of 90 mg/kg showed a statistically significant reduction in the distance traveled by the animals compared to the 0.9% NaCl-treated group (*p* < 0.01), as indicated in Table 1.

*Comparison studies*. In the groups of animals that received IMI-Zn complex at a dose of 20 mg/kg (selected based on dose-effect studies), IMI at a dose of 90 mg/kg, Zn at a dose of 60 mg/kg, and a combination of IMI and Zn at doses of 60 and 40 mg/kg, respectively, there was a significant reduction in the immobility time of mice in both the FST and TST (*p* < 0.0001, *p* < 0.0001, *p* < 0.0001, *p* < 0.05 for FST; *p* < 0.01, *p* < 0.05, *p* < 0.01, *p* < 0.01 for TST). This was determined through statistical analysis using one-way ANOVA followed by Tukey’s post-hoc test compared to the group that received 0.9% NaCl (F(6,63) = 15.42, *p* < 0.0001 for FST; F(6,63) = 6.904, *p* < 0.0001 for TST).

However, in the groups that received IMI at a dose of 60 mg/kg and Zn at a dose of 40 mg/kg, there were no statistically significant changes in the immobility time of mice in either the FST or TST compared to the group receiving 0.9% NaCl (*p* > 0.05).

Importantly, in the group that received the IMI-Zn complex at a dose of 20 mg/kg, the reduction in the immobility time of mice was not statistically different from the reduction observed in the groups that received IMI at 90 mg/kg, Zn at 60 mg/kg, and the combination of IMI and Zn at doses of 60 and 40 mg/kg, respectively (Figure 2).

The tested doses of IMI, Zn, IMI-Zn complex, and joint administration of IMI and Zn did not increase the locomotor activity of mice. In the locomotor activity test, only the group receiving IMI at a dose of 90 mg/kg showed a statistically significant reduction in the distance traveled by the animals compared to the 0.9% NaCl-treated group (*p* < 0.01), as indicated in Table 2.

### 2.2. Biochemical Studies 

In the blood of mice treated with 0.9% NaCl and exposed to severe environmental stress (FST) (group B), no changes in GPX and GR activity, as well as TAS level, were observed as compared to the group of mice not exposed to the stressor (group A) (Table 3). Also, no changes in studied parameters were observed in the mice receiving IMI (60 mg/kg) or Zn (40 and 60 mg/kg) compared to the 0.9% NaCl group. In the mice receiving IMI (60 mg/kg) and Zn (40 mg/kg) simultaneously, no changes in GPX, GR activity, and TAS level were observed compared to the group receiving only IMI (60 mg/kg) or Zn (40 mg/kg). Moreover, IMI-Zn complex at doses of 5, 10, 20, 40, 80, and 100 mg/kg did not influence the GPX, GR activity, and TAS levels as compared to groups of mice receiving IMI alone (60 mg/kg), Zn alone (40 or 60 mg/kg) and IMI (60 mg/kg) and (Zn 40 mg/kg) in combination (Table 3). 

## 3. Discussion

Observations have indicated that by using multiple medications with different but complementary effects, it might be possible to improve the overall response to antidepressant treatment. Moreover, this approach could allow for a more effective antidepressant effect without relying on higher doses of a single medication, which could lead to more adverse side effects [41,42,43,44,45]. The rationale behind polytherapy is to capitalize on the unique mechanisms of action of different substances with antidepressant activity to achieve a synergistic effect. This approach aims to maximize treatment benefits by targeting multiple neurotransmitter systems and modulating various aspects of the complex neurobiology underlying depression [46,47].

Not only antidepressant-antidepressant combinations have been studied for their potential use in treating depression. It was found that several substances can potentiate or enhance the antidepressant activity of commonly used antidepressant drugs. These substances are often called “augmentation agents” or “adjunctive therapies”. These include, among others, atypical antipsychotics (e.g., aripiprazole [48], quetiapine [49], and olanzapine [50]), lithium [51,52], triiodothyronine (T3) [53], omega-3 fatty acids [54], caffeine [55], N-acetylcysteine [56], S-adenosylmethionine [57], as well as essential minerals like magnesium [58], and zinc [59].

Numerous clinical and preclinical studies have demonstrated the antidepressant-like potential of Zn (for review, see [59,60,61,62,63]), prompting investigations into the possibility of enhancing the therapeutic effects of antidepressant drugs through Zn supplementation. Over the past 20 years, researchers have been exploring the potential of Zn to augment the activity of antidepressants with various mechanisms of action. Nowak et al. [64] carried out research demonstrating that the addition of Zn in the treatment of both TCAs and SSRIs led to positive results in easing depressive symptoms among patients with unipolar depression. Further research by Siwek et al. [27] supported these findings, confirming that Zn compounds increase the effectiveness and speed of therapeutic response in treatment-resistant depressed patients receiving IMI. Subsequently, Ranjbar et al. [65] demonstrated that Zn supplementation, when combined with SSRIs, also yields beneficial therapeutic effects. Moreover, preclinical studies have consistently shown the antidepressant effect of Zn, along with the enhancement of antidepressant effects when combined with both inorganic and organic Zn compounds, such as zinc chloride (ZnCl_2_), zinc sulfate (ZnSO_4_), and zinc hydroaspartate [59,60]. Numerous animal tests and models of depression have focused on investigating the effects of co-administering Zn with IMI, a commonly used antidepressant. Various intraperitoneal (i.p.) dosing regimens have been examined, including ZnCl_2_ 30 mg/kg + IMI 20 mg/kg [66], ZnSO_4_ 1 mg/kg + IMI 5 mg/kg [67], zinc hydroaspartate 5 mg/kg + IMI 5 mg/kg, zinc hydroaspartate 10 mg/kg + IMI 15 mg/kg, zinc hydroaspartate 10 mg/kg + IMI 10 mg/kg [68]. These combinations significantly affected rodents’ immobility duration in the FST and TST [66,67,68], indicating their antidepressant-like activity. Likewise, Cunha et al. [69] exhibited a similar outcome in the TST when they administered ZnCl_2_ to mice orally at 1 mg/kg and IMI at 0.1 mg/kg. Furthermore, Rafało-Ulińska et al. [70] reported a significant increase in the activity of mice in the FST after oral administration of subtherapeutic doses of zinc hydroaspartate and IMI at doses of 40 and 60 mg/kg, respectively. The conclusions drawn from the mentioned research, in conjunction with the outcomes presented in this study, suggest that this combination shows promise as an antidepressant therapy while also enabling a reduction in the combined dosage of these substances. This reduction can effectively minimize their adverse effects and expedite the antidepressant response. One of the significant factors contributing to non-compliance in patients is the number of drugs prescribed and the complexity of dosing schedules [71]. When patients are required to take multiple drugs or follow intricate dosing regimens, it can lead to difficulties and contribute to non-compliance and discontinuation of therapy. This issue arises when IMI and Zn are administered as separate oral treatments in animal and human experiments. Therefore, Rogalewicz et al. [72] developed a methodology for obtaining and synthesizing an IMI-Zn complex compound, whose antidepressant activity in the FST in mice was assessed in these studies. The outcomes of the conducted investigation revealed that the IMI-Zn complex displayed an antidepressant-like effect that varied with dosage. This effect was evident through a notable decrease in immobility duration in mice during the FST and TST. Importantly, it is worth noting that the IMI-Zn complex, at any of the concentrations used, did not significantly affect the animals’ motor activity. This finding indicates that the complex exhibited its antidepressant-like effects without causing an increase in locomotor activity. This indicates that the results obtained in despair tests are not false positives.

In the acute toxicity studies conducted using the Zebrafish model, the IMI-Zn complex compound was found to have no adverse effects on fish development. Additionally, it exhibited a dose-dependent impact on the spontaneous locomotor activity of 5-day-old larvae—higher doses of the IMI-Zn complex inhibited locomotor activity, while lower doses increased it [72]. Additionally, Rogalewicz et al. [72] demonstrated that the IMI-Zn complex, specifically at a concentration of 3 μM, produced a similar anxiolytic-like effect in a light/dark test compared to diazepam, a commonly used anxiolytic drug. The authors further showed that the impact of each tested concentration of the IMI-Zn complex differed from those of IMI alone, suggesting that the addition of Zn modified the effect of IMI on the central nervous system [72].

In the present studies aimed at comparing the antidepressant-like activity of the IMI-Zn complex (20 mg/kg) with the activity of single doses of IMI (90 mg/kg) and Zn (60 mg/kg), as well as the combined use of subtherapeutic doses of these substances (60 + 40 mg/kg, respectively), it was observed that the IMI-Zn complex reduced the duration of immobility time of mice in the FST and TST to a slightly lesser extent compared to the groups receiving effective doses of IMI and Zn, as well as the combination of IMI and Zn at subtherapeutic doses. However, these differences were not statistically significant. Consequently, it can be concluded that the antidepressant-like activity of IMI-Zn at 20 mg/kg in mice is comparable to the antidepressant activity of IMI at 90 mg/kg, Zn at 60 mg/kg, and the combined administration of IMI + Zn (60 and 40 mg/kg), respectively. 

Oxidative stress plays a vital role in the pathogenesis of depression, mainly when it is associated with exposure to environmental stress. It has been shown that subjecting rats to chronic stress factors increases protein and lipid peroxidation in brain structures [73,74]. It can lead to neuronal damage and be a significant factor in the development of neurodegenerative diseases. Oxidative stress is a factor that impairs the plasticity of neurons and leads to impaired neurogenesis [75,76]. Therefore, new strategies of pharmacotherapy for these diseases, including depression, should take into account the significant role of oxidative stress in the progeny of conditions [77] and should also take into account the antioxidant properties of substances [74]. A study by Réus et al. [78] showed the antioxidant properties of IMI in an animal stress model. They confirmed that IMI benefits oxidative stress parameters, such as SOD and CAT activity, in the brain structures of animals exposed to acute and chronic stress [79]. Research by Mokoen et al. [79] in an animal model also showed that IMI reverses oxidative damage in the hippocampus induced during an intense stress reaction [79]. However, IMI belongs to TCAs, so it has many side effects. Therefore, despite its considerable effectiveness, it is rarely used in antidepressant therapy. Considering the low efficacy of currently used antidepressants in many cases and the above considerations, in our research, we hypothesized that classic antidepressants in reduced doses, supported by substances with antioxidant potential, may show antidepressant activity based on the mechanisms of the pathogenesis of the disease while maintaining a high safety profile. Doboszewska et al. [80] studied the antioxidant role of Zn. In in vivo studies, Zn-rich and Zn-poor diets were used for comparison. Researchers showed that rats fed a diet rich in Zn had better antioxidant status than animals fed with a diet low in Zn. This proves that Zn has significant antioxidant potential [80]. Also, a study by Emojevwe et al. [81] showed that the administration of Zn ions to rats increased the activity of antioxidant enzymes while reducing ROS and oxidative stress [81]. Therefore, a new approach in the pharmacotherapy of depressive disorders may be the use of antidepressants enriched in Zn compounds. It is worth mentioning that by reducing cellular oxidative stress, Zn has a protective effect on cells [82,83,84].

Therefore, this study aimed to evaluate the parameters of oxidative stress in the blood of mice exposed to an acute stress factor and the IMI-Zn complex administration. Based on the findings from the studies as mentioned above, it is justifiable to assess the antioxidant capabilities of the IMI-Zn complex compound in contrast to the administration of individual drugs. Animals were exposed to FST, which in this study was an inducer of physical stress, leading to increased ROS production, which can then lead to oxidative stress and depressive behavior. Selected oxidative stress parameters were determined to check the antioxidant activity of the substances used. The study showed no statistically significant differences in the measured parameters between mice in the naïve group, which were not subjected to swimming and did not receive any substances, and mice in the control group, which were given 0.9% NaCl solution and subjected to severe environmental stress, such as the forced swimming. This indicates that the stress induced by FST did not lead to oxidative reduction disturbances in the blood of the mice. Although, according to the literature [85,86,87,88,89], FST is a model test of inducing physical stress, no statistically significant changes in the determined oxidative stress parameters were observed in the presented study. It should be noted that the test animals were exposed to the stress stimulus for a short time. Probably the adaptive and defensive abilities of the mouse organism are sufficient to maintain the balance of the redox potential during exposure to a short-stress stimulus. The study should be repeated in the chronic stress model. It has been shown that a 21-day administration of corticosterone to animals leads to depressive symptoms and increased oxidative stress [90].

Based on the literature analysis, it is reasonable to determine the antioxidant potential of the IMI-Zn complex compound compared to drugs administered alone. In this study, animals were administered with IMI or Zn, both alone and in combination, as well as the IMI-Zn complex, to compare the antioxidant potential of the test substances administered alone and in the form of a complex. No statistically significant changes were observed in all parameters determined in the groups of mice receiving the complex compound IMI-Zn (at all doses) compared to the administration of IMI and Zn, either alone or in combination. This could suggest that the antioxidant potential of the complex is comparable to the antioxidant potential of drugs administered alone. However, taking into account the lack of statistically significant changes in the parameters determined in the group of mice subjected to acute stress compared to mice not exposed to this factor, it can be assumed that unambiguous determination and comparison of the antioxidant potential of the tested drugs, administered both individually and in the form of a complex compound, is not possible under these conditions. It can, therefore be assumed that the results obtained in this study may be closely related to the too-short duration of the stress factor. It is also possible, that the lack of statistically significant changes in the determined parameters after administration of the tested drugs to mice may result from their single administration. IMI is a drug whose therapeutic effect is achieved after several weeks of treatment. Considering the literature data confirming the antioxidant activity of both IMI and Zn, as well as the results of behavioral tests of the presented work, it seems reasonable to conduct future studies on the antioxidant status of the IMI-Zn complex in the chronic stress model and with long-term administration of the tested substances.

## 4. Materials and Methods

### 4.1. Animals 

Male Albino Swiss mice (*n* = 310) aged 6–8 weeks were used for the experiment. The mice were obtained from the Experimental Medicine Centre (Lublin, Poland) and housed in standard laboratory conditions (temperature of 21 ± 1 °C, relative humidity of 50 ± 5%) with a 12-h light-dark cycle and ad libitum access to food and water. Before the test, mice were acclimated to the testing room for 1 h. 

All experimental procedures were approved by the Local Ethics Committee (license No. 62/2021) and conducted in line with the guidelines of the European Community Council (2010/63/EU).

### 4.2. Drugs/Substances

The following substances were used in the conducted research: imipramine hydrochloride (IMI, Sigma-Aldrich, Poznań, Poland) at a dose of 60 mg/kg and 90 mg/kg, i.e., 53.096 and 79.644 mg/kg pure IMI (subtherapeutic and effective dose, respectively); zinc hydroaspartate (Zn, Farmapol, Poznań, Poland) at a dose of 40 mg/kg and 80 mg/kg (calculated as pure Zn^2+^ ions; subtherapeutic and effective dose, respectively); imipramine-zinc complex (IMI-Zn, [HL]^2+^[ZnCl_4_]^2−^ (where L: IMI) was synthesized at the Institute of General and Ecological Chemistry, Faculty of Chemistry, Lodz University of Technology, Lodz, Poland [72]. The IMI-Zn complex was administered in the dose range of 5–100 mg/kg (the percentage of Zn in 1 mg of the administered complex is 84.9 µg and 731 µg of IMI). The amount of pure Zn and IMI for individual concentrations of the IMI-Zn complex was presented in Table 4.

All substances were dissolved in 0.9% NaCl immediately before the experiments and were administered orally in a volume of 0.01 mL/g 60 min before behavioral testing. Animals from the control group received 0.9% NaCl. Doses of tested substances and administration schedules were chosen based on our earlier studies and literature data [70,72].

#### Drugs Administration Schedule

I. Dose-effect studies: 1st group: 0.9% NaCl (control group); 2nd group: IMI-Zn complex 5 mg/kg; 3rd group: IMI-Zn complex 10 mg/kg; 4th group: IMI-Zn complex 20 mg/kg; 5th group: IMI-Zn complex 40 mg/kg; 6th group: IMI-Zn complex 80 mg/kg; 7th group: IMI-Zn complex 100 mg/kg; 8th group: IMI 90 mg/kg (positive control group).

II. Comparison studies: 1st group: 0.9% NaCl; 2nd group: IMI 90 mg/kg (effective dose); 3rd group: Zn 60 mg/kg (effective dose); 4th group: IMI 60 mg/kg (subtherapeutic dose); 5th group: Zn 40 mg/kg (subtherapeutic dose); 6th group: IMI + Zn 60 mg/kg + 40 mg/kg (subtherapeutic doses); 7th group: IMI-Zn complex 20 mg/kg (effective dose).

Each experimental group consisted of 10 mice randomly assigned before drug administration.

### 4.3. Behavioural Examination

#### 4.3.1. Forced Swimming Test (FST)

FST was conducted according to the method described by Porsolt et al. [91] with minor modifications. Glass tanks (25 cm height × 10 cm diameter) filled with water (23–25 °C) to a depth of approximately 15 cm were used. Each mouse was separately put into a tank for 6 min. The test session was recorded using a video camera and then analyzed by two blinded investigators. The mice’s immobility time (s) was assessed for the last 4 min of the test duration. An animal was considered immobile when it floated or made only minimal movements to keep the head above water.

#### 4.3.2. Tail Suspension Test (TST)

TST was conducted according to the method described by Steru et al. [92] with minor modifications. Each mouse was gently suspended by the taped tail onto the bar positioned approximately 50 cm above the floor using adhesive tape, ensuring that the mouse was hanging freely without touching any surfaces. The session duration was set at 6 min. The test session was recorded using a video camera and then analyzed by two blinded investigators. The mice’s immobility time (s) was assessed for the last 4 min of the test duration. An animal was considered immobile when it made only minimal movements necessary to breathe.

#### 4.3.3. Locomotor Activity Test

The locomotor activity apparatus (Opto-Varimex-4 Auto-Track, Columbus Instruments, Columbus, OH, USA) consisted of eight square open-field arenas (dimensions: 43 cm × 43 cm × 32 cm) equipped with infrared photocells to detect the mouse’s movement. Each mouse was individually placed in the center of the open-field arena. The session duration was set at 6 min. Locomotor motility was determined as distance traveled (cm) by mice in the last 4 min of the test duration, which corresponds with the time interval evaluated in the FST and TST.

#### 4.3.4. Research Scheme

The research schedule was presented on Figure 1.

### 4.4. Biochemical Analysis

#### 4.4.1. Collection of Blood Samples

Mice were decapitated after performing behavioral tests. The test material was blood collected into an Eppendorf tube filled with EDTA to prevent blood clotting. The samples were then centrifuged at 10,000 rpm for 15 min at 4 °C. After centrifugation, the plasma was collected into empty Eppendorf tubes to obtain material for further studies. Samples were stored at –80 °C for 3 weeks.

#### 4.4.2. Determination of Biochemical Parameters

The prepared samples were tested to determine the activity of oxidative stress parameters. GPX activity, GR activity, and TAS levels were measured using diagnostic kits. GPX and GR activity were measured using the Glutathione Peroxidase Assay Kit and Glutathione Reductase Assay Kit (Cayman Chemical, Ann Arbor, MI, USA). TAS levels were measured using the ImAnOx^®^ kit (TAS/TAC) (Immunodiagnostic AG, Bensheim, Germany).

### 4.5. Statistical Analyze

The recorded data were analyzed using statistical software (GraphPad Prism 10, version 10.0.2) to compare the behavioral parameters between groups. The FST and TST outcomes were shown as mice immobility time (s) arithmetic mean. In contrast, the locomotor activity test is the arithmetic mean of distance traveled (cm) by mice in the last 4 min of the test duration ± standard error of the mean (S.E.M.) for each experimental group.

When applicable, statistical significance was determined using one-way ANOVA and Dunnett’s or Tukey’s post-hoc test, with a significance level set at *p* < 0.05.

## 5. Conclusions

In conclusion, the behavioral studies demonstrate the antidepressant-like potential of the IMI-Zn complex, with its effect shown to be dose-dependent after oral administration to mice. Additionally, findings indicate that the IMI-Zn complex shows antidepressant potential at lower doses (of both Zn and IMI) compared to the single administration of IMI and Zn in therapeutic doses and their joint administration in subtherapeutic doses.

By developing the IMI-Zn complex, it might be possible to simplify the treatment approach, potentially improving patient compliance by combining the therapeutic effects of both IMI and Zn within a single compound, thus addressing one of the contributing factors to non-compliance in depression therapy. The IMI-Zn complex could be a valuable strategy to optimize therapeutic outcomes and balance efficacy and tolerability.

Moreover, it was found that subjecting the mice to an acute stress factor (FST) did not cause statistically significant changes in the activity of GPX, GR, and in the level of TAS, which may indicate too short exposure time of the experimental animals to the stress stimulus. The absence of statistically significant alterations in the measured oxidative stress indicators among mice subjected to the IMI-Zn complex compound could stem from the stress stimulus not disrupting the equilibrium between oxidation and reduction processes. Conversely, the collected data might suggest that the IMI + Zn complex shares comparable antioxidant characteristics with Zn and IMI. It seems reasonable to conduct research using the model of chronic stress and the long-term administration of tested complex and consider the other factors that underly superior properties of the IMI + Zn complex. The study’s constraints involve conducting experiments exclusively on male mice. To substantiate the positive impacts of the Zn-IMI complex, it is imperative to carry out research on female subjects.

## Data Availability

The data presented in this study are available in the article.

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
