# Peer review of "Evaluation of Antidepressive-like Behaviours and Oxidative Stress Parameters in Mice Receiving Imipramine-Zinc Complex Compound"

_ijms, 2023, doi:10.3390/ijms241814157_

Round 1

Reviewer 1 Report

The manuscript "Evaluation of Antidepressive-Like Behaviours and Oxidative Stress Parameters in Mice Receiving Imipramine-Zinc Complex Compound" by Aleksandra Szopa et al. is relevant study and may be of interest to the International Journal of Molecular Sciences readers. In my opinion, the manuscript is a logically constructed study and can be published after taking into account small comments.

- How can the authors explain the phenomenon they found (presented in Table 1), indicating a decrease in motor activity in mice under the action of 90 mg/kg of imine? Could this mean that the animal was in a state of "freezing", which characterizes an increase in the level of animal’s anxiety? In order to confirm or refute this, the authors should also present the results of such an indicator as "time in action". At the moment, the results are somewhat contradictory, since it is absolutely unclear for what reason there was a significant decrease in the distance travelled - it was due to a decrease in movement speed or an increase in stress in mice. Similarly to this remark, the authors should add relevant information to the results of Comparison studies (Table 2).

- Were IMI-Zn dissolved in 0.9% NaCl? In subsection 4.2. Drugs/Substances, the authors should specify in which solvent the studied concentrations of the tested compounds were prepared.

- There are also some typos in the manuscript, in connection with which the authors should carefully check the text. For example, (1) in Figure 2 A and B, the letter "O" in the word "compound" is omitted in the legend. (2) The abbreviation TST is deciphered by the authors only at the 13th mention in the text of the manuscript - you should enter its decryption at the first mention. (3) Line 170 - there is an incomprehensible symbol in the word "naïve". Etc.

As a recommendation, I would like to note that in the future authors should use more modern literary sources. The problem of finding antidepressant drugs has been extremely relevant in recent years, as evidenced by a huge number of experimental and review manuscripts. However, in a study by Szopa et al. the list of sources used mainly consists of studies performed more than 10-15 years ago. During this time, a large number of scientific views have managed to undergo changes and modern points of view are certainly the most reliable due to the existence of improved systems for detecting and a pathological process dynamics.

I would like to wish the authors of the manuscript good luck in future research, and I hope that when lighting their future results they will be able to take my wish into account.

Author Response

We thank both Reviewers for their vulnerable remarks and interest in the presented data. We tried to answer all questions raised in the review. Please see the report below. We have highlighted the changes made in the manuscript in red.

Reviewer 1

  1. - How can the authors explain the phenomenon they found (presented in Table 1), indicating a decrease in motor activity in mice under the action of 90 mg/kg of imine? Could this mean that the animal was in a state of "freezing", which characterizes an increase in the level of animal’s anxiety? In order to confirm or refute this, the authors should also present the results of such an indicator as "time in action". At the moment, the results are somewhat contradictory, since it is absolutely unclear for what reason there was a significant decrease in the distance travelled - it was due to a decrease in movement speed or an increase in stress in mice. Similarly to this remark, the authors should add relevant information to the results of Comparison studies (Table 2).

Ad. 1. Imipramine is a tricyclic antidepressant drug that is primarily used to treat depression and other mood disorders. One of the potential side effects of imipramine is a decrease in locomotor activity in mice. This sedative effect is also observed in humans and can be one of the reasons why imipramine is used cautiously, as it can cause drowsiness and impair motor coordination (Fayez &Gupta, 2023). The exact mechanism by which imipramine causes a decrease in locomotor activity is not fully understood, but it is thought to be related to its effects on neurotransmitter systems in the brain, particularly the serotonin and norepinephrine systems. Imipramine works by inhibiting the reuptake of these neurotransmitters, leading to increased levels of serotonin and norepinephrine in the synaptic clefts between nerve cells. These neurotransmitters play a role in mood regulation, anxiety, and arousal (Liu at al., 2018). The increased levels of serotonin and norepinephrine in the brain can have a sedative effect, which may contribute to the decrease in locomotor activity observed in animals and humans taking imipramine (O’Leary at al., 2007). Additionally, imipramine's effects on other neurotransmitter systems, such as histamine and acetylcholine, could also play a role in its sedative effects (Moraczewski et al., 2023). Over three decades ago, de Felipe et al. 1989 conducted a study where they made an intriguing observation. They found that a dosage of 30 mg/kg of imipramine, a dose recognized for its antidepressant effects in the Forced Swim Test (FST), led to a noteworthy reduction in the motor activity of the experimental animals. However, the 10 mg/kg dosage, considered non-antidepressant in the FST context, did not yield a significant decrease in motor activity. Building upon this initial discovery, subsequent investigations have consistently demonstrated that intraperitoneal and oral administration of imipramine induces a sedative effect. This effect becomes evident through the reduction of spontaneous locomotor activity in the subjects under study ( de Angelis 1996, Poleszak et al., 2006, Pleszak et al., 2016). 

  1. Fayez, R., & Gupta, V. (2023). Imipramine. In StatPearls. StatPearls Publishing.
  2. Liu, Y., Zhao, J., & Guo, W. (2018). Emotional Roles of Mono-Aminergic Neurotransmitters in Major Depressive Disorder and Anxiety Disorders. Front Psychol, 9, 2201. https://doi.org/10.3389/fpsyg.2018.02201
  3. O'Leary, O. F., Bechtholt, A. J., Crowley, J. J., Valentino, R. J., & Lucki, I. (2007). The role of noradrenergic tone in the dorsal raphe nucleus of the mouse in the acute behavioral effects of antidepressant drugs. Eur Neuropsychopharmacol, 17(3), 215–226. https://doi.org/10.1016/j.euroneuro.2006.06.012
  4. Moraczewski, J., Awosika, A. O., & Aedma, K. K. (2023). Tricyclic Antidepressants. In StatPearls. StatPearls Publishing.
  5. de Felipe, M. C., Jiménez, I., Castro, A., & Fuentes, J. A. (1989). Antidepressant action of imipramine and iprindole in mice is enhanced by inhibitors of enkephalin-degrading peptidases. Eur J Pharmacol, 159(2), 175–180. https://doi.org/10.1016/0014-2999(89)90702-4
  6. de Angelis L. (1996). Experimental anxiety and antidepressant drugs: the effects of moclobemide, a selective reversible MAO-A inhibitor, fluoxetine and imipramine in mice. Naunyn Schmiedebergs Arch Pharmacol, 354(3), 379–383. https://doi.org/10.1007/BF00171072
  7. Poleszak, E., Wlaź, P., Kedzierska, E., Nieoczym, D., Wyska, E., Szymura-Oleksiak, J., Fidecka, S., RadziwoÅ„-Zaleska, M., & Nowak, G. (2006). Immobility stress induces depression-like behavior in the forced swim test in mice: effect of magnesium and imipramine. Pharmacol Rep, 58(5), 746–752.
  8. Poleszak, E., Stasiuk, W., Szopa, A., Wyska, E., Serefko, A., Oniszczuk, A., WoÅ›ko, S., ÅšwiÄ…der, K., & Wlaź, P. (2016). Traxoprodil, a selective antagonist of the NR2B subunit of the NMDA receptor, potentiates the antidepressant-like effects of certain antidepressant drugs in the forced swim test in mice. Metab Brain Dis, 31(4), 803–814. https://doi.org/10.1007/s11011-016-9810-5
  9. Were IMI-Zn dissolved in 0.9% NaCl? In subsection 4.2. Drugs/Substances, the authors should specify in which solvent the studied concentrations of the tested compounds were prepared.

Ad. 2. In the subsection 4.2. Drug/Substances: We added information that all substances were dissolved in 0.9% NaCl.

  1. There are also some typos in the manuscript, in connection with which the authors should carefully check the text. For example, (1) in Figure 2 A and B, the letter "O" in the word "compound" is omitted in the legend. (2) The abbreviation TST is deciphered by the authors only at the 13th mention in the text of the manuscript - you should enter its decryption at the first mention. (3) Line 170 - there is an incomprehensible symbol in the word "naïve". Etc.

Ad 3. We have introduced extensions in the first part of the text where they were used and corrected the rest in accordance with this.

  1. As a recommendation, I would like to note that in the future authors should use more modern literary sources. The problem of finding antidepressant drugs has been extremely relevant in recent years, as evidenced by a huge number of experimental and review manuscripts. However, in a study by Szopa et al. the list of sources used mainly consists of studies performed more than 10-15 years ago. During this time, a large number of scientific views have managed to undergo changes and modern points of view are certainly the most reliable due to the existence of improved systems for detecting and a pathological process dynamics.

Ad. 4.  We followed the principle of citing original, source works. Unfortunately, most of them, especially on IMI activity, were published a long time ago.

I would like to wish the authors of the manuscript good luck in future research, and I hope that when lighting their future results they will be able to take my wish into account.

Thank you for your warm wishes, and good luck to you as well.

Author Response

Answer to the Reviewers

We thank both Reviewers for their vulnerable remarks and interest in the presented data. We tried to answer all questions raised in the review. Please see the report below. We have highlighted the changes made in the manuscript in red.

Reviewer 2

  1. In the current study, only male mice were used. The authors should explain why female mice were not included in the study and indicate that the current findings should only be applied to the population of male mice.

Ad. 1. Performing experiments on females is necessary because, as it is known, females do not respond equally as males to treatments or procedures. However, we could not include these studies in these experiments due to limited funding. We will schedule such experiments in our further studies. For now, we mentioned it in the conclusions.

The study's constraints involve conducting experiments exclusively on male mice. To substantiate the positive impacts of the Zn-IMI complex, it is imperative to carry out research on female subjects.

  1. A crucial aspect of the study is selecting the therapy dose. According to the authors, subtherapeutic doses are determined by the outcomes of their prior tests. The cited article did not, however, present or discuss the selected therapeutic doses. In their earlier studies, the authors only applied subtherapeutic dosages. It would be beneficial if the authors of the current study would describe how the optimal doses of Zn and IMI were chosen. If there is no published study reporting data regarding various concentrations of Zn and IMI and their behavioral effectiveness, I advise the authors to state the source of the available data (preliminary results) in the current article as well.

Ad. 2. In selecting the doses of imipramine for our study, we drew upon insights from existing literature and previous research investigations. Specifically, our choice of the 60 mg/kg dose for IMI and 40 mg/kg was guided by previous studies that have identified these dosages as not having antidepressant effects in the context of the FST. Rafało-Ulińska et al. [71] demonstrated that these particular dosages did not lead to significant behavioral changes in mice, whereas their combination caused significant changes in immobility of mice indicative of antidepressant action. Active doses of IMI and Zn (90 and 60 mg/kg, respectively) were chosen based on preliminary studies (unpublished data) conducted in our laboratory.

  1. The final concentrations of prepared solutions of Zn, IMI, and IMI-Zn in physiological saline (?) should also be stated. Were the concentrations of Zn, IMI, or IMI-Zn assessed in the blood of mice after the administration? Were there any differences in the intestinal absorption properties of the IMI-Zn compared to individually or jointly applied treatments? Could this explain why the complex has a higher antidepressant-like potential than other drug delivery methods? Please have some discussion about that

Ad 3. We added information about the content of pure IMI in imipramine hydrochloride and pure IMI and Zn in the IMI-Zn complex for individual doses in the Drugs/Substances subsection.

  1. I suggest the authors stress that their investigation was focused on the exploration of acute drug administration. Also, the timeline of the study is simple and yet not immediately understandable. For instance, I suggest the authors provide behavioral testing time periods not only in the figure caption (Figure 1, line 107) but also in the material and methods sections.

Ad.4. To clarify the research protocol, we added a diagram in the Materials and Methods chapter and a description.

  1. Although there were no changes in the activity of the enzymes, I suggest the authors look into any potential relationships (correlation) between the outcomes of the behavior tests and the activity of the enzymes in order to give more details on their potential connections. It is necessary to include potential (significant) correlations in the article (if any).

Ad.5. Thank you very much for the suggestion to analyze the results in terms of correlations between behavioral effects and markers of oxidative stress. We used Sperman's rho coefficient to identify potential correlations between variables such as outcomes of the TST, FST, and biomarkers such as GPX, GT, and TAS.  The only correlations were found in the subgroups. Strong correlation was found in the following groups: NaCl 0.9% (Control) negative correlation of FST and TAS (r=-0.829) and TST and GPX (r=-0.943); IMI-Zn100 negative correlation of FST and TAS (r=-0.943); IMI-Zn20 positive correlation of TST and TAS (r=0.886); IMI-Zn 40 positive correlation of TST and GPX (r=0.943); IMI-Zn5 negative correlation between TST and GR (r=-0.841); Zn 40 negative correlation of TST and TAS (r=-0.853). However, we are aware of the limitations of our studies and analysis because of the low number of cases (n=6) in subgroups, which may impact the findings. stability and generalizability. In this case, additional data or a larger sample size might be needed to draw more robust conclusions about the relationship in our investigation. We'll pay attention to future studies to make the appropriate cases. Thank you once again for your suggestion.

6a. The authors make the claim that therapeutic effects of antidepressants like IMI are expected after several weeks of administration, which is accurate. However, the behavioral testing conducted on mice after acute therapy administration revealed an immediate positive behavioral outcome, which (according to the evidence) is unrelated to changes or improvements in oxidative stress parameters. How do the authors explain this

Ad. 6a. The exact mechanism of how zinc may exert antidepressant effects is not fully understood, but there are several hypotheses and research findings that suggest a potential role for zinc in the treatment of depression. Proposed mechanisms are neurotransmitter regulation, anti-inflammatory effects, BDNF or mTOR kinase signaling pathway regulation, and antioxidant activity. Obtained results indicated that antioxidant activity might not be the primary mechanism that can underly the more favorable effects of IMI-Zn complex. Thus we will focus our future work on the evaluation of other mentioned above mechanisms.

6b. ? If behavior testing showed antidepressant-like effects of all administered treatments, why would chronic stress exposure and chronic administration of the therapy be needed? Could these aspects be related to time-consuming protein expression changes?

Ad. 6b. Typically, screening tests such as TST or FST are used to evaluate the antidepressant effect of new test compounds. The compounds usually act within a short time after administration in these tests, and the tests are performed on naïve animals. So, in order to validate the antidepressant potential of new compounds, they are also subjected to testing in depression models. In such models, some compounds act after a single administration, and then we conclude that they are fast-acting compounds. Some require longer administration, which is more like the use of drugs in humans. The use of the model would allow to induce of symptoms of depression such as anxiety or memory impairment. Thus using models of depression we will be able to test the anhedonia (key symptom of depression), anxiolytic potential, or the effect on memory using additional tests such as sucrose preference test, elevated plus maze test, novelty suppressed feeding test, or object location test and novel object recognition test respectively.